# Parking Line Based SLAM Approach Using AVM/LiDAR Sensor Fusion for Rapid and Accurate Loop Closing and Parking Space Detection

**DOI:** 10.3390/s19214811

**Published:** 2019-11-05

**Authors:** Gyubeom Im, Minsung Kim, Jaeheung Park

**Affiliations:** 1Graduate School of Convergence Science and Technology, Seoul National University, Seoul 08826, Korea; edwardim@snu.ac.kr (G.I.); minsungkim@snu.ac.kr (M.K.); 2Advanced Institutes of Convergence Technology, Suwon 16229, Korea

**Keywords:** autonomous vehicle, autonomous valet parking, SLAM, around view monitor (AVM), LiDAR, sensor fusion, parking space detection

## Abstract

Parking is a challenging task for autonomous vehicles and requires a centimeter level precision of distance measurement for safe parking at a destination to avoid collisions with nearby vehicles. In order to avoid collisions with parked vehicles while parking, real-time localization performance should be maintained even when loop closing occurs. This study proposes a simultaneous localization and mapping (SLAM) method, using around view monitor (AVM)/light detection and ranging (LiDAR) sensor fusion, that provides rapid loop closing performance. We extract the parking line features by utilizing the sensor fusion data for sparse feature-based pose graph optimization that boosts the loop closing speed. Hence, the proposed method can perform the loop closing within a few milliseconds to compensate for the accumulative errors even in a large-scale outdoor environment, which is much faster than other LiDAR-based SLAM algorithms. Therefore, it easily satisfies real-time localization performance. Furthermore, thanks to the parking line features, the proposed method can detect a parking space by utilizing the accumulated parking lines in the map. The experiment was performed in three outdoor parking lots to validate the localization performance and parking space detection performance. All of the proposed methods can be operated in real-time in a single-CPU environment.

## 1. Introduction

In recent years, autonomous driving technology has been developed rapidly. Many global companies, such as Google, GM, BMW, and Mercedes-Benz are investing heavily in research and development for the commercialization of autonomous vehicles by 2020, and they are aiming for fully autonomous driving by 2025 [1]. The autonomous parking system, which increases the convenience for the driver, is also being actively investigated [2].

Parking is a challenging task for autonomous vehicles. A centimeter level precision of distance measurement is necessary for parking safely at a destination to prevent collisions. Therefore, increasing research is being conducted on parking assist systems (PAS), which have been incorporated in commercial vehicles in recent years, as well as autonomous valet parking (AVP) systems that perform fully autonomous parking [3]. An AVP system is a completely driver-free parking system, in which an autonomous driving vehicle searches for a parking space and then parks itself without a driver’s assistance. For an autonomous vehicle to perform accurate and safe parking by itself, precise and real-time localization technology must be available.

Many researchers have actively studied light detection and ranging (LiDAR) sensor-based simultaneous localization and mapping (SLAM) algorithms to achieve accurate localization performance in outdoor environments [4,5,6]. The LiDAR sensor presents the smallest measurement error among all the range sensors [7]. Therefore, it permits the capture of the minute details of an environment at long ranges. Laser odometry and mapping (LOAM) is a low-drift and real-time LiDAR-based SLAM algorithm [6] that extracts planar/edge features from a LiDAR scan by calculating the roughness of a point in its local region and finding correspondences between scans. The LOAM presents the highest accuracy, and this is achieved by a LiDAR-only estimation method on the KITTI dataset [8]. However, LOAM presents only LiDAR odometry and does not provide loop closure. Errors accumulate over time and cannot be reduced.

To avoid collisions with nearby parked vehicles during the process of parking, the accumulative error must be compensated for. Various studies have been conducted to obtain real-time localization performance in a large-scale outdoor environment [9,10,11]. Hess et al. recently presented Google’s Cartographer SLAM algorithm [11] by aggregating LiDAR data into a local occupancy grid map. Cartographer uses submap-based global optimization in order to achieve real-time performance even in a large-scale map. While mapping, it creates submaps that are linked by constraints in the graph. When loop closure is detected, the positions of submaps are optimized to compensate for the error. However, all LiDAR scans are stored in the grid map over time. The optimization speed reduces gradually and CPU occupancy increases significantly to accelerate the speed. Eventually, the computational time for loop closing takes more than a few seconds.

To tackle this issue of high computational time, several studies have attempted to utilize robust landmarks such as road markings and signs. The matching and storage cost can be significantly reduced by using landmark-based loop closing compared to the feature-based traditional method. Schreiber et al. [12], obtained 3D data using an on-board camera and LiDAR sensors to extract lane information on the road for lane-level localization. Rehder et al. [13] used the road markings obtained by camera images to generate a local grid submap and estimated the position of a vehicle by matching each submap. Jeong et al. [14] extracted the road markings by utilizing the bird’s eye view camera data and classified the markings into six different types of road information to enhance the loop closing performance.

Meanwhile, an around view monitor (AVM) and LiDAR sensor fusion method has been proposed recently for detecting lane markings on complex urban roads accurately [15]. The AVM is a vision sensor system that produces a bird’s-eye view of the 360${}^{\circ }$ surroundings of the vehicle by merging a number of vehicle-mounted cameras. AVMs have been frequently used in parking-related research as they can view all of the directions around the vehicle during parking [16]. However, AVM has a limitation of severe image distortion, which hinders the determination of the position and size of an object. The LiDAR sensor has the smallest measurement error among the ranging sensors [7]. Nonetheless, it is challenging to recognize various road markings in the parking environment using a LiDAR sensor. By fusing an AVM sensor with a LiDAR sensor, we can overcome the disadvantages of individual sensors. The recognition performance can be improved by exploiting the marking recognition by the AVM sensor and the accurate distance measurement by the LiDAR sensor. The study in [15] used the homography matrix calculated by the least squares method [17] to obtain the un-distorted AVM image. Then, the distinct lane markings are extracted from both AVM and LiDAR data to match with the global map. However, this method requires a prior global map created by an accurate real-time kinematic (RTK) GPS sensor to process the map matching with the sensor fusion data.

In this study, we propose a parking line-based SLAM method using an AVM/LiDAR sensor (Figure 1). We employed a LOAM [6] algorithm as the front-end of our method. The parking line features can ensure real-time localization performance even when loop closing occurred in a large-scale map. The parking line feature can be extracted by utilizing the AVM/LiDAR sensor fusion data. Also, thanks to these features, the parking space can be detected by utilizing the parking lines accumulated in the map. All the algorithms operate normally in a single-CPU environment without a GPU. Moreover, these present the advantage of real-time performance. To the best of our knowledge, ours is the first study that utilizes the parking lines to achieve rapid and accurate localization in a parking lot.

The remainder of the paper is organized as follows: Section 2 describes the process of integrating AVM/LiDAR sensors, the process of extracting the parking line feature, and the method of loop closing and detection of a parking space. Section 3 presents the experimental results and discussions. Finally, the paper is summarized and concluded in Section 4.

## 2. Proposed Method

This section describes the procedure of our proposed method. The overall algorithm architecture is shown in Figure 2. The processing pipeline (Appendix A) focuses mainly on extracting the parking line features because it is mandatory for accurate localization and mapping of the proposed SLAM method. First, the AVM and LiDAR sensor are fused as a point cloud and are preprocessed through the sensor fusion module. After the preprocessing process, the AVM data is transformed into a binarized image, and the LiDAR data is transformed into a ground-removed point cloud such as in Figure 2b. Secondly, the parking line feature extraction module filters the AVM obstacle data by using the LiDAR data in order to extract a parking line feature. Then, it registers the sensor fusion data in the map as shown in Figure 2d,e. In this module, principal component analysis (PCA) and histogram analysis are used for extracting the parking line features. Finally, the parking line features are used for both the loop closure module and parking space detection module as shown in Figure 2g,h.

### 2.1. AVM and LiDAR Sensor Fusion

Figure 3 shows that the process of sensor fusion. The AVM sensor data has an image data format of 640×480 pixels and is converted to a point cloud of size 21.2×13.8 m. The actual distance of the AVM image was measured empirically by putting the traffic cones at the edge of the image as shown in Figure 3a. The size of the AVM sensor, 21.2×13.8 m, is a specific value and can be varied depending on the sensor configuration. The LiDAR data has a 3D point cloud data format of [x,y,z]. To maintain an identical dimension to the LiDAR sensor, the AVM data is converted into [x,y,0] format. The converted AVM data is aligned with the LiDAR data based on the LiDAR coordinate system such as in Figure 3b. Axis alignment was performed manually and fifty samples of AVM/LiDAR sensor data were used for accuracy.

Additional preprocessing algorithms are executed on the sensor fusion data to extract a parking line feature. First, the adaptive binarization algorithm is applied to the AVM data in order to obtain the parking line markings in the parking lot. Then, the LiDAR data is transformed into a ground-removed point cloud data by applying Autoware’s space filter [18] as shown in Figure 3c.

The region of interest (ROI) should be set because the distortion of the AVM image hinders the parking line feature extraction process. Thus, the ROI of 0.7 × 2.8 m is empirically set to the area, and only data within this area is used.

### 2.2. Parking Line Feature Extraction

As the parking line feature affects both the loop closing module and the parking space detection module, an accurate parking line feature extraction is essential. This section describes the method for effectively extracting the parking line features with LiDAR-based filtering in a typical urban parking lot environment.

#### 2.2.1. Point Cloud Registration and LiDAR-based Filtering

The sensor fusion data is registered in the map together with the odometry generated using the laser odometry and mapping (LOAM) algorithm. The LOAM is a SLAM algorithm that has an average travel error of 0.59% per meter and a rotation error of 0.0014% (${}^{\circ }$/m) over an 800 m run on the KITTI dataset [8]. When all the 64 channels of data are used as the input for the LOAM, real-time odometry cannot be achieved owing to the computational overload. To maintain the real-time performance, only the 16 channels of the Velodyne sensor data are used as the input.

As the AVM data contains not only the parking lines but also the other binarized data as shown in Figure 4a, the other binarized data should be filtered to extract the parking line features. Hence, the LiDAR data are registered simultaneously with the AVM data to remove the other binarized data from the map. The process of filtering is as follows:Given the LiDAR data, apply the Euclidean clustering algorithm to develop each vehicle cluster.For each vehicle cluster, apply the convex hull algorithm to obtain the boundary points of each cluster.Generate the convex polygons from each convex cluster, as shown in Figure 4c.Then, the AVM data present within the convex polygon are determined as the outliers and erased from the map, as shown in Figure 4d.

Eventually, the LiDAR data is not only used to measure the closest distance but also to filter out the noise data as shown in Figure 4. Both can be possible because of the AVM/LiDAR sensor fusion.

#### 2.2.2. Parking Line Feature Extraction

After acquiring filtered point clouds, we should extract the parking line features. To extract the parking line features exclusively, the registered data is divided into areas of 1.5 × 1.5 m considering the size of one parking space, as shown in Figure 5a. The detailed process is as follows:For each area, apply the Euclidean clustering algorithm to cluster the adjacent data.Apply principal component analysis (PCA) to each cluster to obtain the main axis of the cluster. The main axis is used to obtain the major direction of the parking line.Obtain the data distribution of each cluster along the main axis, as shown in Figure 5c. The data distribution is represented as a histogram.

In order to distinguish the parking line feature from the data in the histogram, we utilized two statistical approaches: the characteristic of the maximum number of scanned points and the kurtosis of the data distribution.

First, the method of the characteristic of the maximum number of scanned points is presented in Figure 6. In the case shown in Figure 6a, the number of scanned points at a certain distance that contains the parking line data tends to be larger than that of the other distance. Here, the maximum number of scanned points of the distance tends to be larger than the sum of the mean μ and two times the standard deviation 2σ. Then, the data corresponding to the distance is obtained as a parking line feature. In contrast, in Figure 6b, the maximum number of scanned points tends to be equal to or smaller than μ+2σ. Then, this data distribution is considered as an outlier, and no corresponding data is extracted as a parking line feature. Secondly, kurtosis is also used as a criterion to extract parking line features. Kurtosis is a statistic component that indicates the data’s sharpness. Kurtosis of the normal distribution is three. When D is a set of data, kurtosis can be obtained as follows:(1)$$\begin{array}{cc} \mathrm{kur}(D) & =E\left[{\left(\frac{D-\mu }{\sigma }\right)}^{4}\right]\;\\ \; & =\frac{E\left(D^{4}\right)-4\mu E\left(D^{3}\right)+6\mu^{2}\sigma^{2}+3\mu^{4}}{\sigma^{4}}, \end{array}$$
where μ is the mean of the data and σ is the standard deviation. $E(D^{3})$ and $E(D^{4})$ represent the third and fourth moments of the data, respectively. As the data distribution of the parking line features is relatively sharper than the other data distribution, kurtosis tends to be higher than that of the normal distribution. Eventually, the histogram satisfying max(D)>μ+2σ and kur(D)>3 is regarded as a parking line feature, and the data of the corresponding max(D) is obtained as a parking line feature.

### 2.3. Parking Line Feature Based Loop Closure

After extracting the parking line features, they are registered in the map as candidates for the loop closure method. As the parking line has a simple repetitive pattern that is likely to fail to identify a correct loop, we also used the point cloud of the parked vehicles with the parking line features, as shown in Figure 7. The detailed procedure of the loop closure module is as follows:(i)Register the parking line feature in the map as a candidate for loop closing along the LiDAR odometry.(ii)When the vehicle revisits a place, the loop closure module attempts to detect a loop inside the distance threshold 5 m.(iii)When a loop is detected, the loop closure module attempts to match the separated parking line features by using the generalized iterative closest point (GICP) [19].(iv)If the matching is successful, the relative pose calculated by the GICP is passed to the general graph optimization (g2o) module to create a loop [20].

The proposed method compensates for the accumulative error by optimizing the following error function:(2)$$\begin{array}{c} x^{*}=\arg\min_{x}\sum_{i}{∥z_{i,i+1}-\hat{z}\left(x_{i},x_{i+1}\right)∥}_{\Sigma_{i,i+1}}^{2}+\sum_{i,j}{∥z_{i,j}-\hat{z}\left(x_{i},x_{j}\right)∥}_{\Sigma_{i,j}}^{2}. \end{array}$$
where $x_{i}$ implies the six degree of freedom (DOF) pose of the vehicle at the i-th node and is defined as follows:(3)$$\begin{array}{c} x_{i}=\left[\begin{array}{cc} R_{i} & t_{i}\\ 0 & 1 \end{array}\right],\;\mathrm{where}\;x_{i}\in \mathrm{SE}\left(3\right). \end{array}$$
$x={\left[\begin{array}{ccccc} x_{1} & \cdots & x_{i} & \cdots & x_{n} \end{array}\right]}^{T}$ represents the vector of *n* vehicle poses. $z_{i,i+1}$ implies the six DOF relative pose between two temporal nodes generated by the LiDAR odometry module. $z_{i,j}$ implies the six DOF relative pose between two non-temporal nodes calculated by the GICP method. The detailed definition of $z_{\left(\cdot ,\cdot \right)}$ is as follows:(4)$$\begin{array}{c} z_{i,i+1}=\left[\begin{array}{cc} R_{i+1}^{i} & t_{i+1}^{i}\\ 0 & 1 \end{array}\right],\;z_{i,j}=\left[\begin{array}{cc} R_{j}^{i} & t_{j}^{i}\\ 0 & 1 \end{array}\right],\;\mathrm{where}\;z_{i,i+1},\;z_{i,j}\in \mathrm{SE}\left(3\right). \end{array}$$

R represents the rotation matrix and t represents the translation vector. $\hat{z}\left(\cdot ,\cdot \right)$ represents the expected measurement of two nodes, and $\Sigma_{\left(\cdot ,\cdot \right)}$ represents the covariance matrix between two nodes. Finally, ${∥e∥}_{\Sigma }^{2}=e^{T}\Sigma^{-1}e$ represents the squared Mahalanobis distance when the covariance matrix is Σ. The illustration of the pose graph SLAM is shown in Figure 7. The black circle and solid line represent the vehicle pose (node) and the constraint (edge), respectively.

### 2.4. Parking Space Detection

The accumulated parking line features in the map are also utilized to detect parking spaces. As the size of the parking space is already determined by the Parking Act, we consider that the size parameter of the parking space is fixed. We assume that the parking spaces in one parking zone share the long-connected line. We proposed the fast and accurate method to detect the parking spaces per each parking zone by utilizing the above assumption.

We categorized the parking line features into two: main line and support line, as shown in Figure 8a. We assume that one parking zone contains a single main line and several support lines perpendicular to the main line. The main line is defined as the long-connected line between each parking space. The support lines are defined as the vertical lines from the main line. In this study, we assumed that the parking lines are rectangular. Thus, we first extract the main line and then sequentially extract the support lines, which are perpendicular to the main line.

#### 2.4.1. Main Line Extraction

The process of extracting the main line is as follows:(i)Apply the region growing cluster algorithm to the parking line features to cluster the parking lines per each separated parking zone.(ii)For each clustered data, apply the PCA again to obtain the major direction of the main line, as shown in Figure 8b.(iii)Obtain the data distribution along the major direction of data, and represent it as a histogram, as shown in Figure 8c.

As the main line has a long tail in a certain direction, the main line can be extracted by returning the distance having the maximum number of scanned points in the histogram. The extracted main line is shown in Figure 8d.

#### 2.4.2. Support Line Extraction

Finally, the location of the support line perpendicular to the main line should be estimated to detect a parking space. The optimal position of the support line can be obtained by optimizing the error using the least squares method, as shown in Figure 9. The following least squares equation can be used to determine the optimal position of the support line:(5)$$\begin{array}{cc} \min_{b}\; & \sum_{p_{i}}∥\frac{\left|ax_{i}-y_{i}+\frac{b_{j}+b_{j+1}}{2}\right|}{\sqrt{a^{2}+1}}∥\\ s.t.\; & a\;⊥\mathrm{Main}\;\mathrm{Line},\\ \; & b_{j}-b_{j+1}=\eta ,\;\mathrm{where}\;j=1,3,5,\cdots \in B,\\ \; & p_{i}\in P,\; \end{array}$$
where *a* represents the slope of the support line perpendicular to the main line and $b_{j}$ represents the intersection of the main line and support line. $p_{i}={\left[x_{i}\;y_{i}\right]}^{T}$ represents a point of a parking line feature. B is the set of b values within the range not exceeding the length of the main line, and P is the set of points located on the main line. As Equation (Equation 5) is a linear regression, a closed-form solution exists. *a* is determined when the main line is determined because the support line is perpendicular to the main line. Therefore, the optimal solution of the $b_{j},b_{j+1}$ can be obtained as follows:(6)$$\begin{array}{c} b^{*}=\frac{b_{j}+b_{j+1}}{2}=\bar{y}-a\bar{x} \end{array}$$
where $\bar{y}$ represents the average of all $y_{i}$ and $\bar{x}$ represents the average of all $x_{i}$. We determine the optimal location of the support line with the minimum error by solving Equation (Equation 6).

When the main line and support line are extracted, the parking spaces can be detected as a combination of these lines, as shown in Figure 8a. Moreover, the occupancy of the parking space can be determined without complex processing. If a convex polygon is present inside a parking space, this parking space is determined as an occupied parking space. Otherwise, it is determined as a vacant parking space.

## 3. Experiments and Results

The experiment was carried out for both SLAM localization performance and parking space detection performance. For the experiments, the autonomous vehicle of Seoul National University Dynamic Robotic Systems (DYROS) Laboratory (shown in Figure 10) was used. The autonomous vehicle platform consisted of a single Velodyne HDL-64E S2 sensor, four AVM cameras, and an X11 RTK GPS sensor. The Velodyne sensor was mounted on the top center of the vehicle, and the AVM cameras were mounted on the front, left, right, and rear of the vehicle. The GPS sensor was mounted behind the Velodyne sensor.

The experiments were conducted in three outdoor parking lots of the Advanced Institute of Convergence Technology located in Gwanggyo-dong, Suwon city (Figure 11). The vehicle moved at an average speed of 8 (km/h) in the parking lot. The PC used in the experiment was an on-board PC (Intel i7-8700, 16 GB of RAM) mounted on the vehicle. Real-time performance was achieved by using a CPU without a GPU. In addition, a robotic operating system (ROS) [21] was used for algorithm development and visualization.

For evaluating the SLAM localization performance, X91 RTK GPS was used as the ground truth. The X91 GPS has an error of approximately 20 (mm) in the fixed state. The results are compared only in the fixed state for accurate experimental results. The absolute translation error (ATE) is used as a performance measuring criterion [22], as shown in Figure 12a. The detailed meaning of the ATE with reference to [22] is as follows:(7)$$\begin{array}{c} \mathrm{ATE}={\left(\frac{1}{N}\sum_{i=0}^{N-1}{∥\Delta p_{i}∥}^{2}\right)}^{\frac{1}{2}}\;\;\mathrm{where}\;\Delta p_{i}=p_{i}-\Delta R_{i}{\hat{p}}_{i}, \end{array}$$
where $p_{i},{\hat{p}}_{i}$ represent the i-th point of the ground truth and the estimated trajectory, respectively. $\Delta p_{i},\Delta R_{i}$ represent the i-th position error and rotation error, respectively.

The detection rate of parking spaces is verified using recall and precision indicators. The accuracy of parking space estimation was verified by comparing the distance error and angular error of the intersection between the actual parking space and the estimated parking space, as shown in Figure 12c.

### 3.1. Parking Line Based SLAM Trajectory Evaluation

For evaluating the localization performance of the proposed SLAM, experiments were conducted on parking lot 1 (PL1), PL2 and PL3. We also conducted experiments using LiDAR sensor-based SLAM, Cartographer [11], and LOAM [6] in order to compare the performance of the proposed method. Cartographer w/ and w/o loop closing (LC) were used for the experiment to compare the effects of loop closure. The ATE [22], which signifies the Euclidean distance between the ground truth data and estimated trajectory, was used, as shown in Figure 12a.

Table 1 presents the result of the experiments performed at the three parking lots. In PL1, the proposed method obtained the smallest root mean square error (RMSE) 1.083 m at a total path length of 471.30 m. Moreover, the proposed method obtained a mean of 1.003 m, median of 0.938 m, and standard deviation of 0.415 m, which are also the smallest among the methods. In case of the maximum and minimum errors, Cartographer w/LC obtained the smallest error with 2.384 m and 0.009 m, respectively.

In PL2-1, the proposed method obtained the smallest RMSE 0.941 m and smallest standard deviation 0.455 m at a total path length of 413.08 m. Moreover, the maximum error of 3.432 m was obtained, which is marginally smaller than the 3.459 m of Cartographer w/LC. However, Cartographer w/LC obtained the smallest mean, median, and minimum errors (0.905 m, 0.853 m, and 0.064 m, respectively).

In the case of Cartographer w/o LC and LOAM without the loop closure process, owing to the accumulative error over time, the maximum errors obtained relatively were large: 3.411 m and 3.814 m in PL1 and 4.386 m and 4.384 m in PL2-1, respectively.

To verify the real-time performance of the proposed SLAM, we compared our result with that of Cartographer W/ Loop Closing to demonstrate the improvement in the computational time. Table 2 presents the comparison result of computational time during loop closing. The proposed method consumed approximately 0.04 [s] to perform the loop closing, whereas Cartographer consumed over 1.3 (s) for both PL1 and PL2-1. A detailed analysis of the experimental results is covered in the Discussion section. The entire results are illustrated in Figure 13 and Figure 14.

### 3.2. Parking Space Detection Performance

The detailed meaning of recall and precision in parking space recognition with reference to [16] is explained in detail as follows:(8)$$\begin{array}{cc} \mathrm{recall} & =\frac{\mathrm{TP}}{\mathrm{TP}+\mathrm{FN}}=\frac{\mathrm{No}.\;\mathrm{of}\;\mathrm{correctly}\;\mathrm{detected}\;\mathrm{spaces}}{\mathrm{No}.\;\mathrm{of}\;\mathrm{existing}\;\mathrm{spaces}}, \end{array}$$
(9)$$\begin{array}{cc} \mathrm{precision} & =\frac{\mathrm{TP}}{\mathrm{TP}+\mathrm{FP}}=\frac{\mathrm{No}.\;\mathrm{of}\;\mathrm{correctly}\;\mathrm{detected}\;\mathrm{spaces}}{\mathrm{No}.\;\mathrm{of}\;\mathrm{detected}\;\mathrm{spaces}}, \end{array}$$where true positive (TP) implies the number of correctly detected parking spaces, false negative (FN) implies the number of missed parking spaces, and false positive (FP) implies the number of incorrectly detected parking spaces. Figure 12b shows the significance of TP, FP, and FN in our results in detail.

Table 3 presents the results of the parking space detection experiments performed at the two parking lots by using the proposed method. The results are the average of five repeated experiments. To ensure diversity of results, the experiments were carried out both in the case of parking spaces being mostly occupied and in the case of being mostly vacant.

In PL1/1, which is a case of fully occupied parking spaces, distinct parking line features could not be extracted owing to the parked vehicle data. A recall of 97.6% and precision of 94.7% were obtained. These values are relatively lower than the recall (97.9%) and precision (97.4%) for PL1/2, which is a case of mostly vacant parking spaces. Similarly, in PL2/1, which is a case of almost occupied parking spaces, a recall of 97.0% and precision of 96.7% were obtained. These are lower than the recall (98.6%) and precision (96.9%) for PL2/2, which is a case of mostly vacant parking spaces.

PL1/3 and PL2/3, which are cases of mostly vacant parking spaces, exhibited the highest detection performance, with recalls of 98.1% and 98.8%, and precisions of 97.6% and 97.4% respectively. Therefore, the recall and precision performance are relatively high in the case of vacant parking space detection. It is considered that it would be effective in practical parking scenarios.

### 3.3. Accuracy of Estimated Parking Space

The accuracy of the estimated parking space was validated by the distance error and angle error between the intersection of the actual parking line and the estimated parking line, as shown in Figure 12c. The intersection represents a cross point between the main line and support line in the parking line. The estimated parking space has been verified only within the ROI area because the AVM sensor data has been used only inside the ROI area.

The experiments were conducted by randomly selecting twenty of the estimated parking spaces and calculating the distance error and angle error. The distance error $e_{t}$ of the two intersections was calculated from the Euclidean distance between the actual intersection and estimated intersection, and the angle error $e_{\theta }$ was calculated from the angle difference between the actual parking line and estimated parking line.
(10)$$\begin{array}{c} e_{t}=∥p_{real}-p_{est}∥, \end{array}$$
(11)$$\begin{array}{c} e_{\theta }=\cos^{-1}\left(\frac{v_{\mathrm{real}}\cdot v_{\mathrm{est}}}{∥v_{real}∥∥v_{est}∥}\right). \end{array}$$

$p_{real}$ represents the actual intersection in the ROI area, and $p_{est}$ represents the estimated intersection. In addition, $v_{real}$ represents the vector in the direction of the parking line at the position $p_{real}$, and $v_{est}$ represents the vector in the direction of the parking line at the position $p_{est}$.

The experimental results reveal that the maximum and minimum distance errors obtained were 0.073 m and 0.057 m, respectively, for PL1; and 0.065 m and 0.045 m, respectively, for PL2 (where the parking lines are relatively distinct). The maximum and minimum angular errors obtained were 0.056 rad and 0.032 rad, respectively, for PL1; and 0.058 rad and 0.038 rad, respectively, for PL2.

### 3.4. Discussion

In the case of the localization, the proposed method was more accurate than LOAM [6] used as a front-end of our method with respect to the ATE even when the autonomous vehicle searched the parking spaces for a long time. Compared to the LOAM, the average RMSE 0.75 m and maximum error 1.1 m were improved. In practical parking scenarios, precision at the centimeter level is mandatory. Therefore, the proposed method is more suitable for a parking scenario than LOAM.

Comparing the performance of the proposed method with Cartographer [11] w/loop closing, there was no remarkable improvement in the accuracy aspect. However, Cartographer consumed over 1.3 s of the maximum computational time to perform the loop closing, whereas the proposed method consumed approximately 0.04 s. The real-time SLAM performance is crucial to achieve parking without colliding with the vehicle parked nearby. Moreover, Cartographer also requires an additional recognition algorithm to detect the parking space. This is because the mapping process of Cartographer was not designed to detect parking spaces. The proposed method is capable of detecting parking spaces by utilizing the accumulated parking lines in the map from the mapping process. Moreover, the proposed method operates in real-time in a single-CPU environment.

In the case of the parking space detection experiment, the average recall and precision obtained were over 97%, which can be considered high. However, the remaining 3% has to be overcome. In order to achieve more reliable performance, the major reasons of the performance degradation need to be discussed. Figure 15 shows the two main factors impacting performance degradation. As shown in Figure 15, the performance deteriorated significantly when the parking lines were concealed by vehicles and when there were shadows.

To improve the recall performance (Figure 15a), a method of predicting the parking space using probability can be considered. The prediction of parking spaces is represented as the probability of actual availability. Moreover, the probability is increased when an actual parking space is available in the adjoining area. In addition, a method that is not affected by outliers can be considered to improve the precision performance (Figure 15b). With the recent developments in deep learning technology, we can train a neural network model that is less influenced by shadows. The precision performance could be increased by utilizing this approach.

## 4. Conclusions

In this study, we propose a parking line-based SLAM approach using AVM/LiDAR sensor fusion for rapid and accurate loop closing and parking space detection. The parking line features are extracted by using the principal component analysis (PCA) and histogram analysis after LiDAR-based filtering and are used for both the localization and mapping process. The parking line-based rapid loop closure method is proposed for accurate localization in the parking lot.

The experimental results reveal that the accuracy of the proposed SLAM was higher than those of two LiDAR-based SLAM algorithms. Moreover, the computational time for loop closing is significantly shorter than that of the Cartographer SLAM algorithm, and real-time performance can be achieved.

For the mapping aspect, the accumulated parking line features in the map are classified as either the main line or support line to detect parking spaces. Then, the parking space is detected as a combination of the main line and support line. The experiments reveal that the average recall and precision are 98% and 97%, respectively. Furthermore, the average distance error and rotation error of the estimated parking space are 0.05 m and 0.03 rad, respectively.

The proposed method presents the limitation that only data within the region of interest (ROI) can be measured so dynamic obstacles cannot be detected. However, as we mentioned in the section concerning obstacle filtering, we can utilize the LiDAR data to measure the closest distance to a dynamic obstacle by generating convex polygons. Moreover, the proposed method works only where there are parking lines in the parking lot, and it considers only the rectangle-shaped parking spaces. Our future work is aimed at developing a parking space detection method for parking spaces of various shapes.

## Figures and Tables

**Figure 1 sensors-19-04811-f001:**
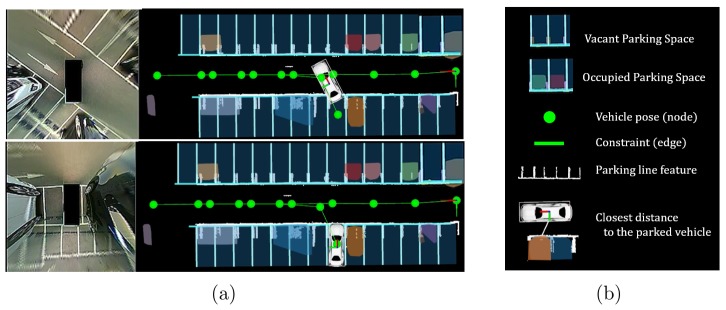
Illustration of the proposed method. The parking line features are extracted from the around view monitor (AVM)/light detection and ranging (LiDAR) sensor fusion data. These are used for both rapid loop closing and detection of parking spaces. (**a**) The left figures show the actual AVM images, and the right figures show the proposed method. (**b**) Graphical explanation of each symbol.

**Figure 2 sensors-19-04811-f002:**
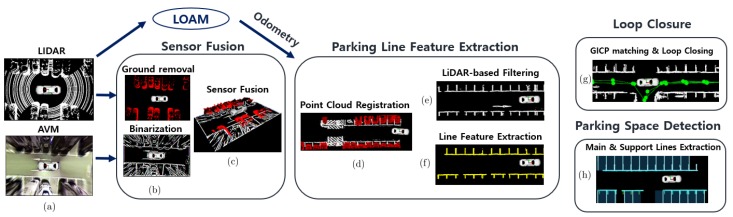
Process pipeline of the whole algorithm. Each module is executed sequentially.

**Figure 3 sensors-19-04811-f003:**
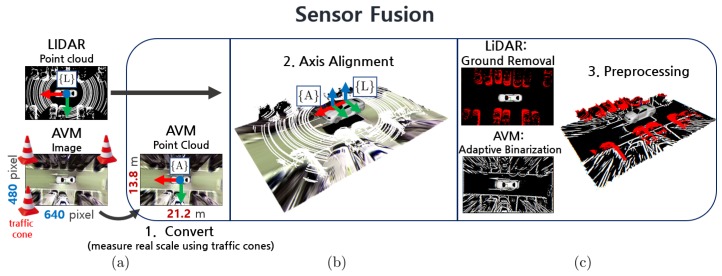
Process of sensor fusion. (**a**) Convert AVM image to point cloud. (**b**) Align axis of both LiDAR and AVM sensor data. (**c**) Preprocessing sensor data to extract a parking line feature.

**Figure 4 sensors-19-04811-f004:**
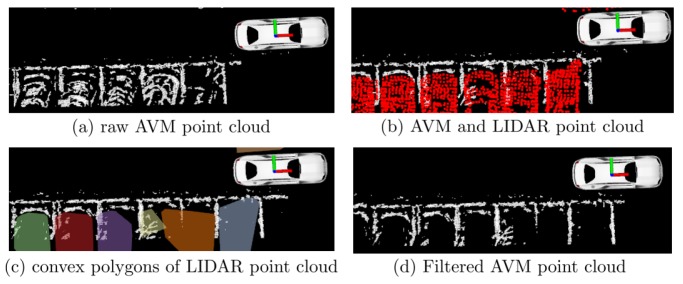
The process of LiDAR-based filtering to extract parking line feature. (**a**) Raw AVM point cloud without filtering the parked vehicle data. (**b**) The AVM and LiDAR data are registered in the map. (**c**) Convex polygons are generated by applying a convex hull to the obstacle data. (**d**) Filtered point cloud.

**Figure 5 sensors-19-04811-f005:**
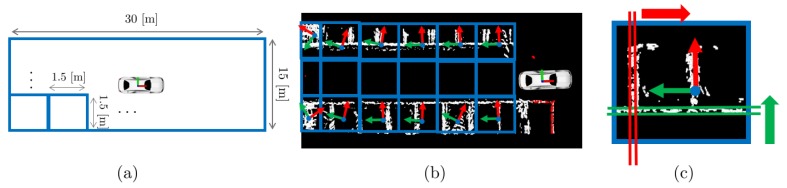
The data registered in the map is cut into areas of 1.5 × 1.5 m (blue box), and the principal component analysis (PCA) is applied to each area. (**a**) The dimension of division areas. (**b**) The PCA is used for obtaining the major axis of the data for each area (red and green arrows). (**c**) Determination of data distribution in the direction along the major axis for analyzing the data characteristics.

**Figure 6 sensors-19-04811-f006:**
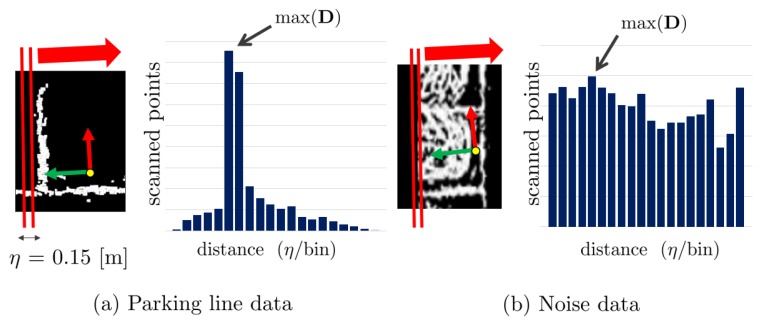
Process of extracting parking line features. The histogram illustrates the data distribution in the vertical direction of the main axis (red arrow). (**a**) Histogram distribution of parking line data. (**b**) Histogram distribution of data with noise. For (**b**), the parking line feature is not extracted.

**Figure 7 sensors-19-04811-f007:**
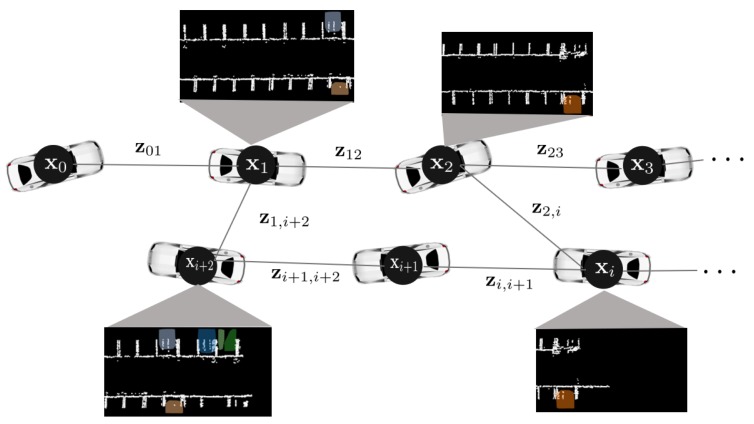
Illustration of pose graph. Each node represents a six degree of freedom (DOF) vehicle pose. The edge is the constraint between two nodes. The temporal edges are generated by LiDAR odometry, and the non-temporal edges are generated by the generalized iterative closest point (GICP) method. All the edges are corrected to compensate for the error that accumulated when the loop closing occurred.

**Figure 8 sensors-19-04811-f008:**
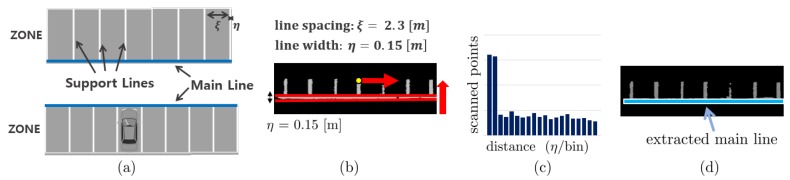
(**a**) Graphical image of parking spaces. Both the main line and support line are defined arbitrarily. The thickness of the support line is already specified as η=0.15 m, and the size of the parking space is specified as ξ=2.3 m in our parking environment. The parking line specification is determined by the Korean Parking Act and can be changed according to the country or region. (**b**) Scan the data distribution of the main line. (**c**) The histogram shows the data distribution of the main line. The bin of the main line tends to be longer than those of the other locations. (**d**) The main line is extracted by selecting the maximum bin in the histogram.

**Figure 9 sensors-19-04811-f009:**
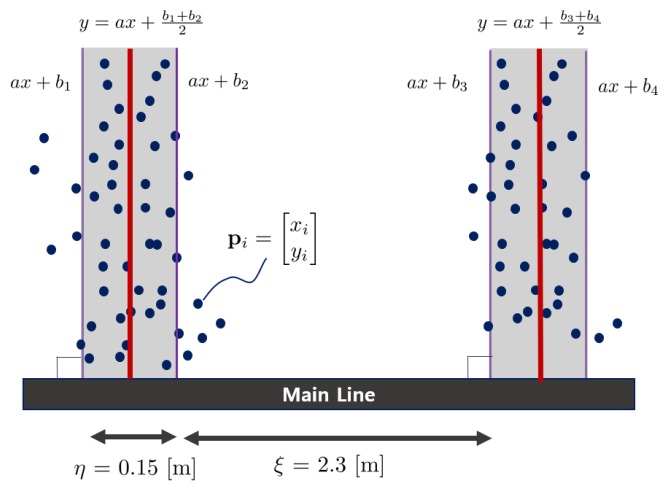
The process of identifying the support line. The support line can be estimated by optimizing the least squares method. The red line indicates the center of the support line.

**Figure 10 sensors-19-04811-f010:**
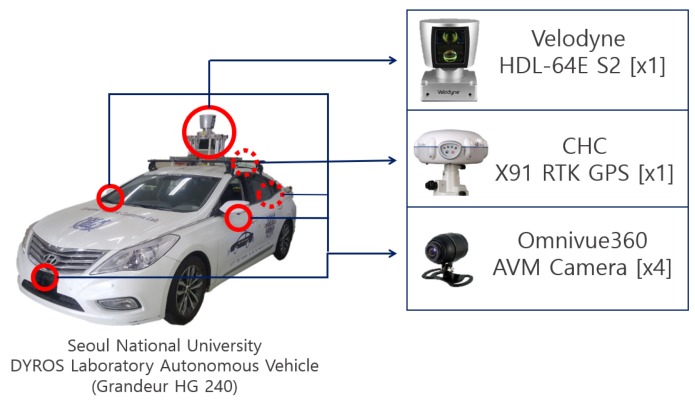
Sensor configuration of the autonomous vehicle. The autonomous vehicle of Seoul National University Dynamic Robotic System (DYROS) Laboratory was used.

**Figure 11 sensors-19-04811-f011:**
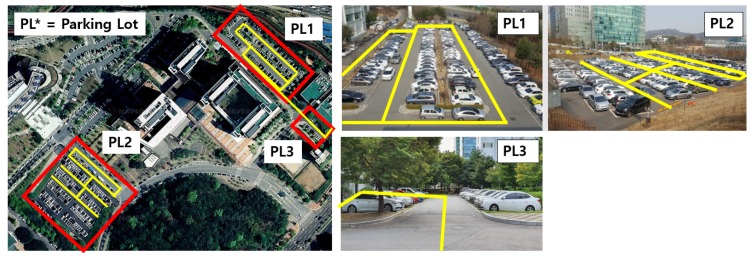
Three outdoor parking lots at the Advanced Institute of Convergence Technology. The yellow lines indicate the path that the autonomous vehicle passed for evaluating the simultaneous localization and mapping (SLAM) performance.

**Figure 12 sensors-19-04811-f012:**
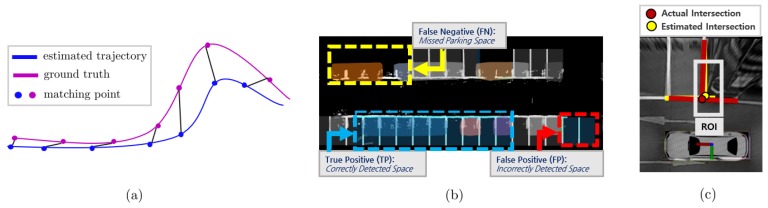
Details of performance measuring criteria for SLAM trajectory evaluation and parking space detection. (**a**) Process of evaluating absolute trajectory error (ATE). (**b**) Meanings of true positive (TP), false negative (FN) and false positive (FP) in our experiment. (**c**) Actual intersection and estimated intersection for measuring accuracy of parking space.

**Figure 13 sensors-19-04811-f013:**
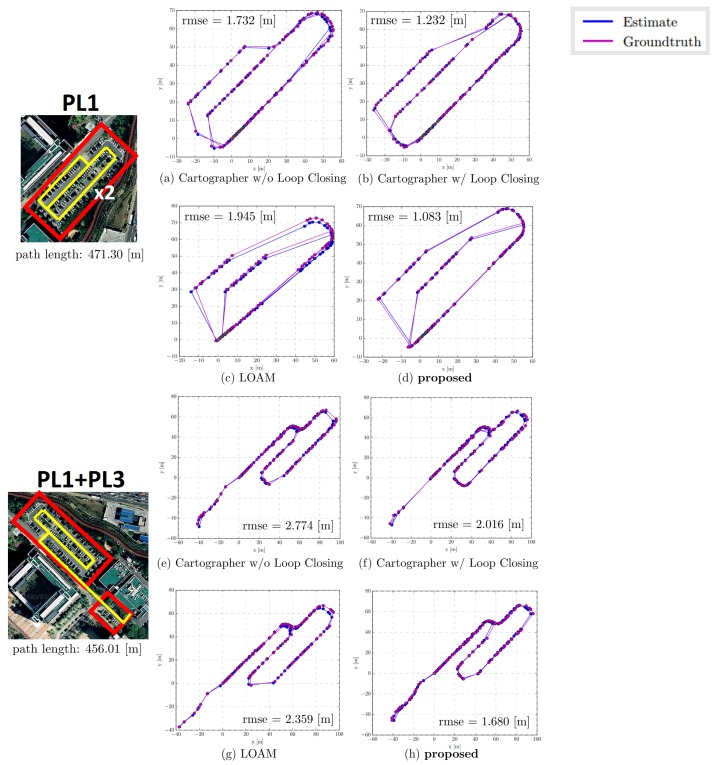
Illustration of SLAM trajectory evaluation. Experiments were conducted in two outdoor parking lots PL1 and PL1+PL3. The evaluation was performed by comparing between the ground truth (GPS) and estimated trajectory. The blue line indicates the estimated trajectory generated by the SLAM algorithm, and the purple line indicates the ground truth (GPS). The blue and purple dots indicate the location where timestamp matching occurred. In the case of (**c**,**d**), the timestamp matching occurs sparsely, and the path appears relatively sharp.

**Figure 14 sensors-19-04811-f014:**
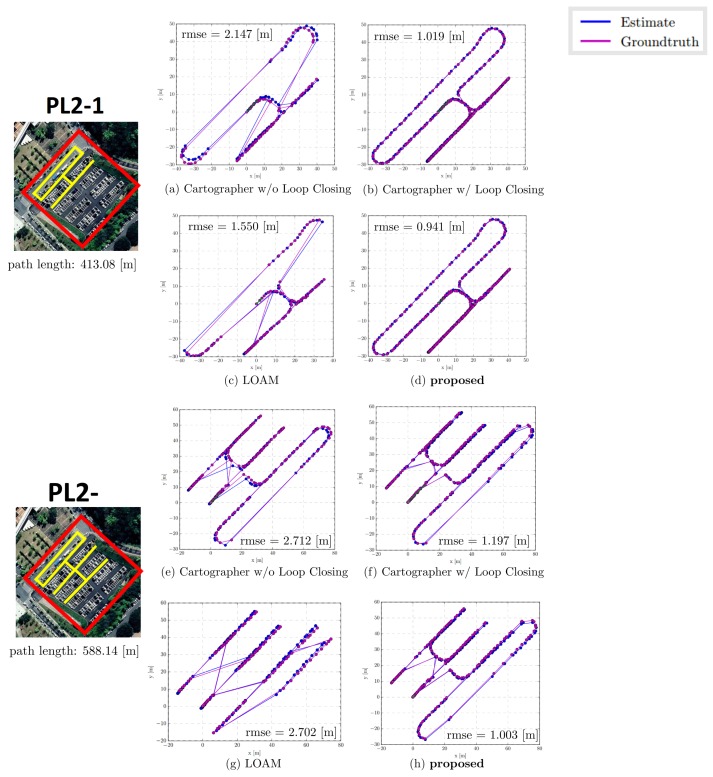
Illustration of SLAM trajectory evaluation. Experiments were conducted in the outdoor parking lot PL2-1 and PL2-2. In the case of (**c**,**g**,**h**), the timestamp matching occurs sparsely, and the path appears relatively sharp.

**Figure 15 sensors-19-04811-f015:**
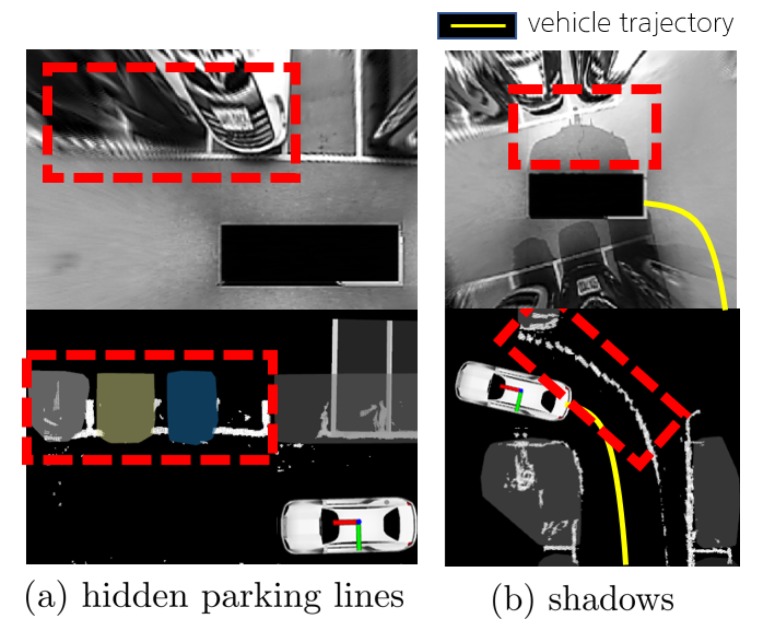
The main reasons for parking space detection performance degradation. (**a**) The case where the parking line is obstructed by vehicles. (**b**) The case where shadows are accumulated along the trajectory of the vehicle.

**Table 1 sensors-19-04811-t001:** Experimental result of absolute trajectory error. Parking lot (PL), loop closing (LC), standard deviation (STD), root mean square error (RMSE). Bold values mean the smallest value achieved among the methods.

SLAM Method	Case	Max (m)	Mean (m)	Median (m)	Min (m)	STD (m)	RMSE (m)
Cartographer w/o LC	PL1	3.411	1.502	1.162	0.442	0.862	1.732
Cartographer w/LC	PL1	**2.384**	1.07	1.044	**0.009**	0.610	1.232
LOAM	PL1	3.814	1.655	1.567	0.132	1.022	1.945
**proposed**	PL1	2.491	**1.003**	**0.938**	0.013	**0.415**	**1.083**
Cartographer w/o LC	PL1+PL3	6.477	2.296	2.206	0.204	1.556	2.774
Cartographer w/LC	PL1+PL3	5.199	1.712	1.557	**0.008**	1.05	2.016
LOAM	PL1+PL3	6.756	1.825	1.407	0.108	1.461	2.359
**proposed**	PL1+PL3	**3.911**	**1.369**	**1.054**	0.030	**0.974**	**1.680**
Cartographer w/o LC	PL2-1	4.386	1.856	1.622	0.274	1.080	2.147
Cartographer w/LC	PL2-1	3.459	0.905	0.853	0.064	0.469	1.019
LOAM	PL2-1	4.384	1.280	1.149	**0.013**	0.874	1.550
**proposed**	PL2-1	**3.432**	**0.823**	**0.769**	0.083	**0.455**	**0.941**
Cartographer w/o LC	PL2-2	7.613	2.285	2.319	0.047	1.461	2.712
Cartographer w/LC	PL2-2	2.705	1.076	0.944	0.035	0.525	1.197
LOAM	PL2-2	8.400	2.143	1.813	0.118	1.645	2.702
**proposed**	PL2-2	**2.659**	**0.897**	**0.891**	**0.005**	**0.447**	**1.003**

**Table 2 sensors-19-04811-t002:** Computational time comparison for loop closing (LC) process.

SLAM Method	Case	Total # of LC Occurred	Max LC Time Spent (s/LC)
Cartographer w/LC	PL1	3	1.5312
**proposed**	PL1	16	0.0412
Cartographer w/LC	PL2-1	4	1.3229
**proposed**	PL2-1	17	0.0381

**Table 3 sensors-19-04811-t003:** Experiment result of parking space detection using proposed method.

Parking Lot/Trial	# of Spaces	# of Occupied Spaces	# of Vacant Spaces	Average TP + FP	Average TP	Recall (%)	Precision (%)	Average Distance Error (m)	Average Rotation Error (rad)
PL1/1	141	141	0	145.4	137.8	97.6%	94.7%	0.073	0.032
PL1/2	145	10	135	145.8	142.0	97.9%	97.4%	0.065	0.056
PL1/3	42	5	37	42.2	41.2	98.1%	97.6%	0.057	0.047
PL2/1	54	52	2	54.2	52.4	97.0%	96.7%	0.065	0.038
PL2/2	57	14	43	58.0	56.2	98.6%	96.9%	0.058	0.039
PL2/3	52	0	52	52.8	51.4	98.8%	97.4%	0.045	0.058

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
