# Peer review of "Parking Line Based SLAM Approach Using AVM/LiDAR Sensor Fusion for Rapid and Accurate Loop Closing and Parking Space Detection"

_sensors, 2019, doi:10.3390/s19214811_

Round 1
Reviewer 1 Report
The authors present a procedure to detect lines in outdoor parkings and the free space in them. Although the proposal might be interesting, it is necessary for the authors to modify it.
The authors say that "... proposes a novel simultaneous localization and mapping (SLAM) method". However, it is not clear which is the new SLAM proposal that is proposed. The SLAM algorithm employed is SLAM [19]. It is necessary to clarify this aspect.
It would be convenient to redo the manuscript, indicating precisely and clearly what are the main contributions. For example, in the first section (Introduction), the global idea is repeated in excess but it is not clear which are the main contributions in relation to other proposals.
Although the procedure as a whole might be interesting, it is not clear what the novelty of the proposal is, since it consists of a succession of procedures that a priori are not entirely new. The main proposal focuses on a parking line detection system. In this sense, the relationship with obstacle detection is not understood (it could be two parallel procedures).
The proposed method uses information from a vision system and a laser system (Velodyne). Except for some details in which the information provided by the laser is used to eliminate areas of the image, the fusion of both systems is not sufficiently justified.
Claims are made that are not entirely true. For example, "The AVM has been frequently used in parking-related research because it is equipped on a large number of commercial vehicles". This statement is not true today.
How is aligned the image data with the laser data?
The parking line feature extraction, that is the main procedure, is unclear. For example, "the detailed process is as follows: 1. For each area, apply ..." How the data are divided in areas?
To detect if the vehicle has revisited a place, is only the information provided by the GPS used?
Only the proposed procedure is evaluated in two outdoor parkings. More experiments should be proportioned.
Minor questions:
The acronym AVM is employed. It is curious that when the acronym SLAM is used, which is perfectly known, its meaning "simultaneous localization and mapping (SLAM)" is indicated, but when using the acronym AVM, which is much less known and employed, its meaning is not indicated The image provided by the AVM sensor has 640x480 pixels. What is the meaning of 'converted to a point cloud of size 21.2 x 13.8 [m]'.
Author Response
Please see the attachment for more detailed revision history.
Point 1: The authors say that "... proposes a novel simultaneous localization and mapping (SLAM) method". However, it is not clear which is the new SLAM proposal that is proposed. The SLAM algorithm employed is SLAM [19]. It is necessary to clarify this aspect.
Response 1: We have removed 'novel' word and clarified that we employed SLAM [19] as a front-end of our method in Introduction section.
Point 2: It would be convenient to redo the manuscript, indicating precisely and clearly what are the main contributions. For example, in the first section (Introduction), the global idea is repeated in excess but it is not clear which are the main contributions in relation to other proposals.
Response 2: We indicate three main contributions more clearly in Introduction section. The proposed method has the following three contributions.
1. Fast and accurate parking line-based loop closing can be performed even when an autonomous vehicle has been searching the parking lot for a long time.
2. As a result of the mapping process, accurate parking spaces can be detected by utilizing the accumulated parking lines in the map.
3. All the algorithms operate normally in a single-CPU environment without a GPU. Moreover, these have the advantage of real-time performance.
Point 3: The main proposal focuses on a parking line detection system. In this sense, the relationship with obstacle detection is not understood (it could be two parallel procedures).
Response 3: As a result of the mapping process in proposed SLAM, the AVM/LiDAR sensor fusion data is accumulated in the map. Thus, we can detect the parking lines from AVM data and measure the closest distance to the vehicle nearby (obstacle) from LiDAR data. The LiDAR data is not only used to measure the closest distance but also filter out the noise data as shown in Figure 4. Both can be possible because of the AVM/LiDAR sensor fusion.
We have added the above explanation in 'Obstacle Filtering' section (2.2.1).
Point 4: The proposed method uses information from a vision system and a laser system (Velodyne). Except for some details in which the information provided by the laser is used to eliminate areas of the image, the fusion of both systems is not sufficiently justified.
Response 4: After the LiDAR data is accumulated in the map as shown in Figure 4(b), we can develop each vehicle cluster by utilizing the Euclidean clustering algorithm. For each cluster, apply convex hull algorithm to generate the convex polygons of each cluster as shown in Figure 4(c). Eventually, the AVM point cloud data present within the convex polygon are determined as the outlier and eliminated from the map as shown in Figure 4(d).
We have added Figure 3 to explain the detail procedure of sensor fusion.
Point 5: How is aligned the image data with the laser data?
Response 5: We have aligned the converted AVM point cloud(x,y,0) with LiDAR point cloud(x,y,z) only for [x,y] manually. Fifty samples of AVM/LiDAR sensor data were used for this alignment. We have added the above explanation in 'Sensor Fusion' section (2.1) and also added the visualized information in Figure 3.
Point 6: The parking line feature extraction, that is the main procedure, is unclear. For example, "the detailed process is as follows: 1. For each area, apply ..." How the data are divided in areas?
Response 6: We have added more detailed information about the division of area in Figure 5. The area of 30 x 15 [m] is divided by 1.5 [m] squares to extract the parking line features.
Point 7: To detect if the vehicle has revisited a place, is only the information provided by the GPS used?
Response 7: The LOAM [19] odometry algorithm is used to detect if the vehicle has revisited a place. Once the odometry is overlapped in the distance threshold 5 [m], the loop detection algorithm is executed. GPS data isn't used for the loop detection.
Point 8: Only the proposed procedure is evaluated in two outdoor parkings. More experiments should be proportioned.
Response 8: We conducted two more experiments (PL1+PL3, PL2-2 as shown in Figure 13, 14 and Table 1.
Point 9: The acronym AVM is employed. It is curious that when the acronym SLAM is used, which is perfectly known, its meaning "simultaneous localization and mapping (SLAM)" is indicated, but when using the acronym AVM, which is much less known and employed, its meaning is not indicated.
Response 9: We have indicated the meaning of the acronym AVM (around view monitor).
Point 10: The image provided by the AVM sensor has 640x480 pixels. What is the meaning of 'converted to a point cloud of size 21.2 x 13.8 [m]'?
Response 10: The traffic cones were placed at the end of the AVM image 640x480 [pixel] and the actual distance 21.2 x 13.8 [m] was measured manually as shown in Figure 3. We have added more explanation in 'AVM and LiDAR Sensor Fusion' section.

Reviewer 2 Report
The paper is well written and I have little comments.
Some abbreviations are used which for the general readers needs to be explained, examples are AVM, ego-vehicle, CNN.
For readability figure 4 (and maybe 3 could be better on the next page since it is only referenced there).
In black/white most of the images are not readable, but that is difficult to solve.
The explanation of the combination algorithms used lengthy but detailed. It could add readability to signpost back to the overall structure at regular intervals.
In the experiments, explain that PL is parking lot.
Are the two experiments sufficient to validate your algorithm, support this assumption in text. Why these two tracks?
In table 2 i do not get the reason why they are bold? wrong rows?
Argument more what the impact is of a precision of e.g. 94.7% does this mean that in 5.3% of the cases it will park where there is already a car? Is this corrected when it drives closer?
You argue that sometimes the lines are not detected since there are cars, what happens in parking lots where there are no lines but other ways of mapping out the spots?
Author Response
Please see the attachment for more detailed revision history.
Point 1: Some abbreviations are used which for the general readers needs to be explained, examples are AVM, ego-vehicle, CNN.
Response 1: I have corrected the terms that were pointed out by the reviewer 2. AVM to around view monitor (AVM), ego-vehicle to the vehicle, CNN to convolutional neural network (CNN).
Point 2: For readability figure 4 (and maybe 3 could be better on the next page since it is only referenced there).
Response 2: We moved Figure 3,4 (now 4,5) on the next page for readability.
Point 3: The explanation of the combination algorithms used lengthy but detailed. It could add readability to signpost back to the overall structure at regular intervals.
Response 3: We have modified the explanation of loop closure and parking line detection using itemize for readability.
Point 4: In the experiments, explain that PL is parking lot.
Response 4: We have added the explanation 'PL = Parking Lot' in Figure 11.
Point 5: Are the two experiments sufficient to validate your algorithm, support this assumption in text. Why these two tracks?
Response 5: We have conducted two more experiments (PL1+PL3, PL2-2) as shown in Figure 13, 14 and Table 1.
Point 6: In table 2 i do not get the reason why they are bold? wrong rows?
Response 6: Thank you for pointing that out. The wrong bold values are corrected.
Point 7: Argument more what the impact is of a precision of e.g. 94.7% does this mean that in 5.3% of the cases it will park where there is already a car? Is this corrected when it drives closer?
Response 7: The precision of 94.7% means that there is 5.3% chance that the empty space will be misdetected as parking space. The detailed definition is as follows: Precision = No. correctly detected spaces / No. detected spaces. The precision is improved when the vehicle drives closer.
We have mentioned the above explanation in 'Parking Space Detection Performance' section (3.2).
Point 8: You argue that sometimes the lines are not detected since there are cars, what happens in parking lots where there are no lines but other ways of mapping out the spots?
Response 8: Currently, the proposed method cannot detect a parking space without parking line features. We have added the explanation in Conclusion section.

Reviewer 3 Report
Dear authors,
I have just two remarks to your paper.
1.) The platform where the algorithms were tested was not described. Is this a self driving car? I can see / read about the sensors, but there is no mention of the self driving car.
2.) Safety. The system has some blind spots / limitations that should be noted in the paper. Especially what happens in terms of dynamic obstacles. Was this considered?
Author Response
Please see the attachment for more detailed revision history.
Point 1: The platform where the algorithms were tested was not described. Is this a self driving car? I can see / read about the sensors, but there is no mention of the self driving car.
Response 1: The vehicle we used for the experiment was a self driving car. The self driving car of Seoul National University DYROS Laboratory was used(model: HYUNDAI Grandeur HG 240). We added information about self driving car in Figure 10.
Point 2: Safety. The system has some blind spots / limitations that should be noted in the paper. Especially what happens in terms of dynamic obstacles. Was this considered?
Response 2: We have added the limitation of our method in Conclusion section. We didn't consider the dynamic object in the proposed method. However\, we can utilize the LiDAR data to measure the distance to the dynamic obstacle by generating a convex polygon as we mentioned in obstacle filtering section.

Round 2
Reviewer 1 Report
The authors have lightly justified the questions previously raised in the first revision. However the main questions remain unaltered.
The abstract has not been modified The introduction section the global idea is repeated in excess but it is not clear which are the main contributions in relation to other proposals. The paper needs a complete rewrite.
Author Response
Please see the attachment for more revision details.
Point 1: The abstract has not been modified The introduction section the global idea is repeated in excess but it is not clear which are the main contributions in relation to other proposals. The paper needs a complete rewrite.
Response 1:
We have completely rewritten the introduction section and heavily revised the abstract and conclusion section to highlight our main contribution. We also revised the title of the paper to 'Parking Line-based SLAM using AVM/LIDAR Sensor Fusion for Rapid and Accurate Loop Closing and Parking Space Detection'.
Our main contribution is the rapid loop closing performance by utilizing the parking line features, in a parking lot. In order to avoid collision with parked vehicle nearby while parking, real-time localization performance should be maintained even when loop closing occurred. Because of the sparse characteristic of the parking line features, we can perform the rapid loop closing within a few milliseconds to compensate for the accumulative errors during navigation.
